# ProofAug: Efficient Neural Theorem Proving via Fine-grained Proof Structure Analysis

**Haoxiong Liu** [1]   **Jiacheng Sun** [2]   **Zhenguo Li** [2]   **Andrew C Yao** [1 3]

## Abstract

The synergy between deep learning models and traditional automation tools, such as built-in tactics of the proof assistant and off-the-shelf automated theorem provers, plays a crucial role in developing robust and efficient neural theorem provers (NTPs). However, for proof synthesis with LLMs, previous work applies automation tools either only when explicitly invoked by the model or at a single granularity level, failing to fully exploit their power. To solve this issue, we propose *ProofAug*, a procedure that equips LLMs with automation methods at various granularities through fine-grained structure analysis of model-generated proof proposals. ProofAug also serves as a versatile plug-and-play module that seamlessly integrates with any tree-search algorithm, enabling our construction of an efficient recursive proving (ERP) module to further enhance performance. The superiority of our method is validated on the miniF2F benchmark using the open-source deepseek-math-7b-base model and the Isabelle proof assistant. Notably, by additionally employing a mixed prompting strategy, we achieve a cumulative pass rate of 66.0% after curation of the dataset (61.9% for the original version) with at most 2100 queries to the model per problem (In contrast, the previous SOTA in Isabelle, Subgoal-XL (Zhao et al., 2024), only achieves 56.1% using 16384 queries per problem). We also implement a Lean 4 version of ProofAug that can improve the pass@1 performance of Kimina-Prover-Preview-Distill-1.5B from 44.3% to 50.4% on miniF2F-test. Our code is available at https://github.com/haoxiongliu/ProofAug.

[1]Institute for Interdisciplinary Information Sciences, Tsinghua University [2]Huawei Noah's Ark Lab [3]Shanghai Qi Zhi Institute. Correspondence to: Haoxiong Liu <liuhx20@mails.tsinghua.edu.cn>, Zhenguo Li <li.zhenguo@huawei.com>.

*Proceedings of the $42^{nd}$ International Conference on Machine Learning*, Vancouver, Canada. PMLR 267, 2025. Copyright 2025 by the author(s).

## 1. Introduction

Automated theorem proving is a field that not only appeals to mathematicians looking for efficient formalization and proof automation for mathematical theorems, but also finds significant applications in real-world industries such as integrated circuit design and software engineering (Bibel, 2013; Schumann, 2013). Early applications of machine learning methods on automated theorem proving focus on tasks such as premise selection (Irving et al., 2016) and proof search (Loos et al., 2017) in SMT solvers or resolution/-superposition provers, often interleaved with handcrafted heuristics. Recently, rapid progress in mathematical reasoning with large language models (LLMs) has awoken a flourishing interest for directly synthesizing formal proofs using generative language models.

In literature, there are primarily two paradigms for direct proof synthesis: the proof-step generation paradigm and the whole-proof generation paradigm, each corresponding to a distinct proof style. For proof assistants where proofs are mainly in the procedural style (or tactic-style), such as MetaMath (Megill & Wheeler, 2019) and HOL Light (Harrison, 2009), previous work adopts the proof-step generation paradigm (Polu & Sutskever, 2020; Lample et al., 2022; Han et al., 2022; Polu et al., 2022), where in each iteration the model is requested to generate one tactic to be applied in the interactive theorem prover (ITP)[1]. Then the ITP provides information on the updated proof state, which sometimes could be indispensable for people to finish the remaining proof. Proof-step generation methods typically use tree-search algorithms to guide the proof process, inspired by the success of AlphaGo (Silver et al., 2016). In contrast, for proof systems where declarative style proofs are common, such as Isabelle and Lean 4, computation-efficient whole-proof generation methods become possible. However, the efficiency in this paradigm is not free. Any incorrect application of a proof method that bridges the gap between two proof terms (or goals/conjectures) will cause the proof to fail to pass the ITP verification check.

---

[1]We use the terms 'interactive theorem prover' and 'proof assistant' interchangeably with a nuance: the former emphasizes the formal system aspects, while the latter highlights its role in assisting users.

To avoid this issue, the Draft, Sketch, and Prove (DSP) framework (Jiang et al., 2023) proposes to let the model first write a formal proof sketch (Wiedijk, 2004) according to an informal proof draft, then offload the proofs of intermediate conjectures to off-the-shelf ATPs[2] and heuristic proof methods. Many subsequent works (Wang et al., 2023a; Zhao et al., 2023; Zheng et al., 2024; Zhao et al., 2024) also follow the practice of DSP when designing their few-shot prompts: The proof methods in the original proofs of the demonstration examples are replaced with a placeholder, which signifies that ATPs are responsible for completing the task. We observe that while this approach can significantly enhance performance compared to direct full-proof generation, it may also result in proof failures in unintended ways:

- (Hard Conjectures) In DSP, the in-context learning ability of LLMs makes them follow the style of the few-shot prompt to generate rough sketches according to the informal proof, refusing to dive into the necessary details. As a result, the generated intermediate conjectures could be too hard for ATPs to solve.

- (Complicated Draft) Due to common inconsistencies between informal proof and formal system, there are conjectures that despite being easy to be solved by ATPs, humans need to write complicated informal proofs for them. The unnecessary intermediate steps could be translated into formal conjectures that are inappropriate in syntax or even harder than the original theorem.

Here, a dilemma appears. On one hand, instructing the model to produce a rough sketch increases the risk of missing critical details, making it challenging for ATPs to bridge the gaps. On the other hand, directing the model to generate highly detailed sketches can exacerbate formalization issues.[3] Neither way is ideal for fully leveraging the power of off-the-shelf automation methods.

In this paper, we propose to address this dilemma through performing fine-grained structural analysis on the proof proposals generated by LLMs, based on the observation that regardless of correctness, full proofs contain rich structural information from which valid *semi-proofs*[4] of varying granularities can be derived. Specifically, given a theorem statement, instead of instructing the model to generate a proof sketch as previous work does, we choose to start from generating a full proof to suppress the 'Hard Conjectures'

issue in the first place. For the generated proof proposal, we first find its corresponding *maximal compatible semi-proof* (MCSP) that maintains most structural information yet can pass the check of ITP. Then, departing from it, we recursively resort to a more coarse semi-proof whenever ATPs fail to fill in some gap in the current one. This procedure solves the 'Complicated Draft' issue, thus significantly improving the sample efficiency. We call this procedure *ProofAug*. Figure 1 shows an example flow of our method.

We also demonstrate that the application of proof structure analysis is not restricted to single-pass generation. In fact, ProofAug can be incorporated into any tree-search algorithm to expand the candidate node set at each iteration. This will become evident in our unified view of theorem proving introduced in Section 2.1. In Section 3.3, we illustrate its application in constructing a sample-efficient recursive proving (Wang et al., 2024) module on top of ProofAug.

We list our contributions with corresponding key evaluation results below. All experiments use the open-source deepseek-math-7b-base model (Shao et al., 2024) if not otherwise specified.

- We propose a versatile plug-and-play procedure ProofAug that can significantly enhance the performance of neural theorem provers, based on fine-grained proof structure analysis. On miniF2F-test[5] (Zheng et al., 2021), ProofAug achieves a pass rate of 52.5% with no more than 100 queries to the model. Ablation results show that when compared with the DSP baseline, the improvement brought by ProofAug implementation for Isabelle is 7.8%, 4.1% and 3.3% under sample budgets of 1, 10 and 100, respectively.

- ProofAug can be incorporated into any tree-search algorithms under our unified view of existing generative LM based theorem proving methods. We showcase that when integrated with our Efficient Recursive Proving (ERP) module, 0-shot ProofAug performance improves from 54.5% to 56.1% with a limit of 500 queries per problem. For comparison, RMaxTS (Xin et al., 2024b) only shows an advantage of $< 0.2\%$ over naive single-pass retrying under a sample budget of 3200.

- Moreover, using a mixture of strategies with different prompting methods, we achieve a total pass rate of 61.9% within 1700 model queries. After curating the miniF2F-test data, we manage to prove 66.0% of the theorems with no more than 2100 requests, setting a new state-of-the-art across methods using the Isabelle proof assistant.

- We also build a Lean implementation of ProofAug and

---

[2]For simplicity, we follow the practice of Jiang et al. (2023) to use the abbreviation ATP to refer to any automated method, including SMT solvers and what traditionally known as automated theorem provers.

[3]We refer the readers to Appendix B for two concrete examples that illustrate these two issues.

[4]By *semi-proofs*, we refer to proofs that possibly contain symbols indicating pending proof, such as **sorry** in Isabelle and Lean.

[5]We use 'miniF2F-test' to refer to the miniF2F test split Isabelle data included in the code repository of Jiang et al. (2023). See Section 4.1 for a detailed discussion on the versions of miniF2F.

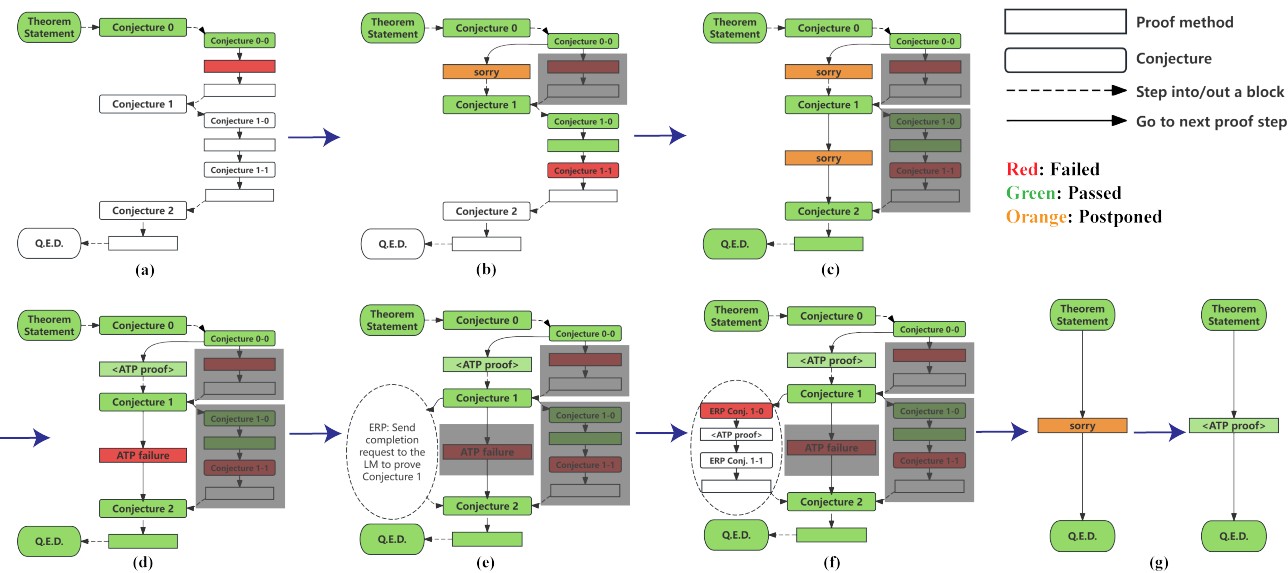

Figure 1: **Illustration example of ProofAug.** Each box in the flowchart corresponds to a proof step. **(a)** The initial proof encounters an error when proving Conjecture 1-0. **(b)** We replace the original proof of Conjecture 1-0 with a **sorry** and continue, until a syntactic error occurs at Conjecture 1-1. **(c)** The proof of Conjecture 1 is replaced by **sorry** and we obtain a semi-proof with two **sorry**s. **(d)** Automation tools are called to prove Conjecture 0 and Conjecture 1, but failing at the latter one. **(e)** The ERP module (introduced in Section 3.3) sends a completion request at Conjecture 1. **(f)** The generated ERP proof still fails at Conjecture 1. **(g)** We finally resort to a more coarse level of proof and successfully find a proof with automation tools. Refer to Algorithm 2 for a pseudo-code description of ProofAug.

shows that the pass@1 accuracy of Kimina-Prover-Preview-Distill-1.5B on miniF2F-test can be improved from 44.3% to 50.4%, exhibiting the generality of ProofAug for various proof systems.

## 2. Preliminaries

### 2.1. A Unified View of Theorem Proving with Generative Language Models

We first formulate the task of theorem proving in the context of language modeling and introduce our notations. We assume there is a global alphabet $\Sigma$. An ITP is formulated as a triple $(\mathcal{A}, \mathcal{S}, T)$, where:

- $\mathcal{A} \subset \Sigma^{*}$[6] is the set of *proof steps*. There is also a parser $\mathrm{Parse} : \Sigma^* \to \mathcal{A}^*$ that transforms a string into a sequence of proof steps.
- $\mathcal{S}$ is the set of *states*. A state $s \in \mathcal{S}$ represents our abstraction of the proof object underlying the ITP, which contains all information the ITP needs to proceed with the proof, including $s$.state, $s$.error, $s$.finish, etc., varying depending on the specific ITP environment. Additionally, there is a special element $s_0$, which represents the initial state.

---

[6] $\mathcal{A}^*$ denotes the set of all finite lists (or sequences) of elements from $\mathcal{A}$, including the empty list.

- $T : \mathcal{S} \times \mathcal{A} \to \mathcal{S}$ is the state transition function of the ITP.

For any string $x \in \Sigma^*$, we use $s_x$ to denote $T(s_0, x)$, where we have lifted the notation $T$ to express consecutive state transitions, i.e., for $s \in \mathcal{S}$,

$$T(s, x) := T(\cdot, \mathbf{a}[n]) \circ T(\cdot, \mathbf{a}[n-1]) \circ \cdots \circ T(\cdot, \mathbf{a}[1])(s)$$

if $\mathrm{Parse}(x) = (\mathbf{a}[1], \mathbf{a}[2], \cdots, \mathbf{a}[n])$.

Given a theorem statement (with context) $x_f \in \Sigma^*$, we say $y_f \in \Sigma^*$ is a *proof* of $x_f$ if $s_{x_f \| y_f}$.finish = True, where $\|$ denotes the concatenation of two strings. A generative language model (with a decoding strategy) is a map that outputs a distribution $\pi(\cdot|x)$ on $\Sigma^*$ when conditioned on a *prompt* $x \in \Sigma^*$. A *response* is a sample $y \sim \pi(\cdot|x)$. The task of theorem proving is to find a proof $y_f$ of the given theorem statement $x_f$ using the black-box distribution generator $\pi$. Some benchmarks also provide a corresponding informal statement $x_i$ and informal proof $y_i$ for each theorem.

We state our unified view of existing generative-LM-based theorem proving methods as follows. In each round, an algorithm completes three steps:

1. (*State Selection*) Select a state to work with in this round.
2. (*Prompt Formation*) Form a prompt and present it to

Table 1: A summary of current theorem proving methods using generative language modeling under our unified view, including DSP (Jiang et al., 2023), LEGO-Prover (Wang et al., 2023a), Lyra (Zheng et al., 2024), GPT-f (Polu & Sutskever, 2020), HTPS (Lample et al., 2022), ReProver (Yang et al., 2023), Lean-STaR (Lin et al., 2024), POETRY (Wang et al., 2024), RMaxTS (Xin et al., 2024b). We use the terms *node* and *state* interchangeably.

| Method | State Selection | Prompt Components | Expected Output | Response Processing |
|---|---|---|---|---|
| *Single-pass generation methods* | | | | |
| Naive | Always $s_{x_f}$ | $x_f$ | a full proof | verify the proof in ITP |
| DSP | Always $s_{x_f}$ | $x_f$ + informal draft | proof sketch | call ATPs to fill the gaps |
| LEGO | Always $s_{x_f}$ | $x_f$ + draft + skills | skill comple. + proof sketch | call ATPs + update skill library |
| Lyra | Always $s_{x_f}$ | $x_f$ + last attempt error | proof sketch | tool correction + call ATPs |
| *Proof-step generation methods* | | | | |
| GPT-f | max logprob goal | the selected goal | a tactic | expand the goal-based search tree |
| HTPS | max PUCT value goal | the selected goal | a tactic | expand and backprop the hypertree |
| ReProver | max logprob state | retrieved prem. + ps | a tactic | expand the state-based search tree |
| Lean-STaR | current state | proof state | a proof step with rationale | proceed the step in ITP |
| POETRY | max logprob state | context + proof state | a proof step (with 'sorry') | update the recursive proof tree |
| RMaxTS | discounted UCB | context + proof state | proof completion | add a new node for each valid step |
| *Ours* | | | | |
| ProofAug | Always $s_{x_f}$ | $x_f$ + informal draft | a full proof | FindMCSP + Proof Augmentation |
| + ERP | the first ATP failure | context + proof state | a proof of this conjecture | + new nodes for each block |

the model.

3. (*Response Processing*) Use the response of the model to interact with ITP to generate new states.

Compared to typical views of tree-search algorithms, our approach integrates their expansion step and back-propagation step with the process of interacting with the ITP into a single response processing module. We isolate the prompt formation procedure into a single module to emphasize its importance, as proof-step generation methods and whole-proof generation methods differ fundamentally at this point. Besides, this allows us to place not only these two lines of work, but also methods that lie in between, under our unified view.

Table 1 summarizes and compares different methods under this view. Our view disentangles the different modules of algorithm design, naturally motivating us to optimize existing algorithms in the module-level. Our method *ProofAug*, which will be introduced in Section 3.2, is a procedure belonging to the response processing step. Thus, we can integrate effective practices in other modules with ProofAug to make further improvements. We showcase such an application in Section 3.3. Moreover, since the goal of theorem proving is to improve the overall pass rate rather than the success rate of each attempt, an appropriate 'portfolio' of different strategies for each module could significantly improve the pass rate than a single strategy. Such a phenomenon has been observed in the CoT & non-CoT mode combination (Xin et al., 2024b; Zhao et al., 2024). In Section 4.4, we explore this idea across additional combinations

```
theorem "((A ⟶ B) ⟶ A) ⟶ A"
proof
  assume "(A ⟶ B) ⟶ A"
  show A
  proof (rule classical)
    assume "¬ A"
    have "A ⟶ B"
    proof
      assume A
      with ⟨¬ A⟩ show B by contradiction
    qed
    with ⟨(A ⟶ B) ⟶ A⟩ show A ..
  qed
qed
```

Figure 2: An example proof in Isar with hierarchical proof...qed block structures.

and demonstrate its superiority.

## 2.2. The Isabelle/Isar Proof Language

Although the basic idea of our method in general fits any ITP, the implementation details vary depending on specific features of the proof assistant. In this paper, we use Isabelle/Isar (Wenzel et al., 2004) as an example to introduce the spirit of ProofAug and how to implement it in a specific proof system.[7] In this section, we introduce some basic knowledge and unique features that ease the implementation

---

[7]See Appendix C for a detailed explanation of why our method is generic across different systems and a brief summary of our Lean implementation.

of our method in Isar.

The specific object corresponding to our 'state' abstraction formulated in Section 2.1 is Isar/VM transition. The transitions are typed with different modes, including **proof(prove)**, **proof(state)** and **proof(chain)**. We find that identifying the mode **proof(prove)** is particularly useful for the analysis of proof structure since it indicates that a goal has just been declared and the next proof step is required to contain a proof method. In Isar, proof methods are only allowed to be invoked after the keywords **proof** and **qed**. As a result, a typical proof in Isar is organized by hierarchical **proof**...**qed** blocks.

Figure 2 shows an example of Isar proof. Users are allowed to **have** or **show** are keywords that Isabelle allows to appear in the **proof(state)** mode to introduce new local goals. To prove a goal with some proof method $m$, one can use a single clause '**by** $m$', which is an abbreviation of '**proof** $m$ **qed**'. The other keywords, **assume** and **with**, serve exactly the same purpose as they do when used in natural language proofs. Benefiting from the above properties, Isar proofs are readable even for people unfamiliar with Isabelle, thus easy to be analyzed just like how we analyze the structure of an informal proof.

Finally, we specify the attributes we can access from a state $s$ through the ITP environment used in this work. We can have a proof state string $s$.state that corresponds to the 'proof state' Isabelle shows to the user, a corresponding mode $s$.mode, a list of all available facts $s$.facts that can be obtained by the **print_facts** command in Isabelle, and two boolean-type attributes $s$.error and $s$.finish indicating whether 'an error has occurred' and whether 'a theorem is proved', respectively. We say $s_1 = s_2$ for two states $s_1, s_2 \in \mathcal{S}$ if all the values of these five attributes are equal.

## 3. Method

Algorithm 2 describes our method, ProofAug. It mainly consists of three parts: finding the Maximal Compatible Semi-Proof (MCSP), proof augmentation, and an optional Efficient Recursive Proving (ERP) module. In this section, we state each of them in detail. For an illustrative walk-through example of ProofAug, refer to Figure 1.

### 3.1. Find the Maximal Compatible Semi-Proof

Given a proof proposal $y_f^0$ sampled from $\pi(\cdot | p(x_i \| y_i, x_f))$ (where $p(\cdot, \cdot)$ is a few-shot/zero-shot prompter that appropriately combines its arguments), the first procedure of ProofAug is to find the *Maximal Compatible Semi-Proof* (MCSP) of $y_f^0$.

We first explain what the MCSP is. Recall that a *semi-proof* refers to a *proof* that possibly contains **sorry** steps. We say a

---

**Algorithm 1** Find the Maximal Compatible Semi-Proof

**Input:** initial proof $y_f^0$, ITP $(\mathcal{A}, \mathcal{S}, T)$
$\mathbf{a} \leftarrow \text{Parse}(y_f^0)$
$\mathbf{s} \leftarrow [\text{Null}] \times \text{len}(\mathbf{a})$      ▷ States before each step
$i, s_{this} \leftarrow 1, s_0$
**while** $i \leq \text{len}(\mathbf{a})$ **do**
    $\mathbf{s}[i] \leftarrow s_{this}$
    $s_{next} \leftarrow T(s_{this}, \mathbf{a}[i])$
    **if** $s_{next}.error$ **then**
        **if** $s_{this}.mode = \text{proof(prove)}$ **then**
            $\mathbf{a}[i] \leftarrow \text{sorry}$
        **else**      ▷ Error in other modes, skip the block
            $block \leftarrow \text{InnermostBlock}(i, \mathbf{a})$
            **if** $block$ **is** Null **then**
                **return** Null  ▷ $\mathbf{a}[i]$ not in any block, terminate
            **end if**
            $\mathbf{a}[block.start..(block.end - 1)] \leftarrow \text{Null}$
            $\mathbf{a}[block.end] \leftarrow \text{sorry}$
            $i, s_{this} \leftarrow block.end, \mathbf{s}[block.start]$
        **end if**
    **else**
        $i, s_{this} \leftarrow i + 1, s_{next}$
    **end if**
**end while**
**if** $s_{this}.finish$ **then**
    **return** $\text{Concat}(\mathbf{a})$
**end if**

---

semi-proof $y$ is *compatible* to $y_f^0$ if each **sorry** in $y$ matches a **proof**...**qed** block or a **by** clause in $y_f^0$ and the remaining parts are equal. Thus, by MCSP, we refer to the particular compatible semi-proof $y_f^M$ from which all compatible semi-proofs (w.r.t. $y_f^0$) can be obtained by substituting some **proof**...**qed** blocks or **by** clauses of $y_f^M$ with **sorry**s.

The FindMCSP routine is described in Algorithm 1. Firstly, $y_f^0$ is parsed into a sequence of proof steps $\mathbf{a}$. Then we create a pointer $i$ indicating the current position and an array $\mathbf{s}$ indicating the states before the $i$th step. In each iteration, we first try to proceed the current proof step $\mathbf{a}[i]$ from the current state $\mathbf{s}[i]$ in the ITP. If this step succeeds, we continue to the next proof step; otherwise, according to the proof mode of $\mathbf{s}[i]$, we take different strategies: if $\mathbf{s}[i]$.mode is **proof(prove)**, we replace $\mathbf{a}[i]$ with **sorry**; otherwise, we find the innermost **proof**...**qed** block of $y_f^0$ that contains the $i$-th step and substitute the whole block with a single **sorry** step, then move the pointer accordingly. The algorithm terminates when a failed step occurs outmost or all steps pass the check of the ITP.

**Algorithm 2** Proof Augmentation (ProofAug)

---

**Input:** theorem statement $x_f$, informal statement & draft $x_i \| y_i$, prompter $p(\cdot, \cdot)$, LM $\pi(\cdot|\cdot)$, ITP $(\mathcal{A}, \mathcal{S}, T)$

$y_f^0 \sim \pi(\cdot|p(x_i\|y_i, x_f))$      ▷ Sample an initial proof

$\mathbf{a} \leftarrow \text{Parse}(\text{FindMCSP}(y_f^0))$      ▷ Apply Algorithm 1

$\mathbf{s} \leftarrow [\text{Null}] \times \text{len}(\mathbf{a})$

$i, s_{this} \leftarrow 1, s_0$

**while** $i \leq \text{len}(\mathbf{a})$ **do**

    $s_{next} \leftarrow T(s_{this}, \mathbf{a}[i])$

    **if** $\mathbf{a}[i] \neq$ sorry **then**

        $error \leftarrow False$

    **else**

        $error \leftarrow T(s_{this}, \texttt{<ATP>}).error$      ▷ Try ATPs

    **end if**

    **if** $error$ **and** $useERP$ **then**      ▷ ERP module

        $y_f^p \leftarrow \mathbf{a}[1..i-1]\|s_{this}.state$

        $y_f^c \sim \pi(\cdot|p(x_i\|y_i, x_f\|y_f^p))$

        **if** $T(s_{this}, y_f^c) = s_{next}$ **then**

            $\mathbf{a}[i], error \leftarrow y_f^c, False$

        **else**

            $y_f^a \leftarrow \text{FailedTactics2ATP}(y_f^c)$

            **if** $T(s_{this}, y_f^a) = s_{next}$ **then**

                $\mathbf{a}[i], error \leftarrow y_f^a, False$

            **end if**

        **end if**

    **end if**

    **if** $error$ **then**      ▷ Resort to the last level

        $block \leftarrow \text{InnermostBlock}(i, \mathbf{a})$

        **if** $block$ **is** Null **then return** Null

        $\mathbf{a}[block.start..(block.end-1)] \leftarrow \text{Null}$

        $\mathbf{a}[block.end] \leftarrow$ sorry

        $i, s_{this} \leftarrow block.end, \mathbf{s}[block.start]$

    **else**

        $i, s_{this} \leftarrow i+1, s_{next}$

    **end if**

**end while**

**return** Concat($\mathbf{a}$)      ▷ The final proof

---

### 3.2. Proof Augmentation

The goal of ProofAug is to determine whether there exists a compatible semi-proof that can be completed into a valid proof using ATPs or built-in heuristic proof methods to fill the **sorry** gaps. Instead of first finding out all compatible semi-proofs and trying them sequentially, we should design a more efficient way that makes use of the overlap among these semi-proofs to avoid waste of computation resources spent on calling ATPs. The resulting proof augmentation algorithm is shown in Algorithm 2 (Ignore the ERP module for now).

Specifically, we first parse the MCSP and create a pointer and a corresponding state array as in Algorithm 1. When we encounter a **sorry** step as we progress with the proof, we try a predefined set of Isabelle built-in heuristic methods and Sledgehammer to find a proof of the current goal. This procedure can be seen as a special proof method `<ATP>` created by us, similar to the built-in proof method **try**. If this `<ATP>` method fails, we can rule out the possibility that the final proof contains the innermost **proof**...**qed** block in which this **sorry** lives and replace the block with **sorry**. We iteratively perform this operation until we finish the proof, or the semi-proof degrades into a single **sorry** step and `<ATP>` fails to directly prove the original theorem.

For a concrete example of ProofAug inducing a proof from a failed initial proof proposal, refer to Appendix I.

### 3.3. Efficient Recursive Proving

Solving a complicated theorem from a single proof proposal generated by the language model $\pi(\cdot|\cdot)$ is challenging even with our ProofAug procedure. By simply sampling more proof proposals, we can probably get better results as long as there is some randomness underlying the decoding strategy of $\pi$. We ask the question: how can we outperform this naive retrying strategy, given a fixed sample budget?

On miniF2F, Wang et al. (2023a); Zheng et al. (2024) achieve better performance than the naive approach by making use of the previous error messages or proved lemmas in the previous attempts of proofs. Nevertheless, these methods all belong to the whole-proof generation paradigm and benefit from the strong instruction-following ability of GPT-series models. Here, we are more interested in going beyond whole-proof generation to investigate whether tree-search methods used in the proof-step generation paradigm can help improve the sample efficiency.

To the best of the authors' knowledge, RMaxTS(Xin et al., 2024b) is the only successful attempt to integrate some tree-search algorithm with whole-proof generation. However, the advantage of RMaxTS only becomes clear when the sample budget comes to $4 \times 6400$. Our explanation for this inefficiency is that, for proofs that deviate from the correct paths at a very early stage, RMaxTS could spend much time in searching after the proof steps that have already misled the direction of proof. We argue that this can be largely avoided if we can first check whether the proof proposal forms a valid reasoning path. POETRY(Wang et al., 2024) is a top-down approach that satisfies this property through fine-tuning the model to never step into the next proof level. However, POETRY falls in the proof-step generation paradigm and requires fine-tuning the model to follow their special proof-step generation strategy.

In order to achieve a more sample efficient and fine-tuning free recursive proving method, we design our algorithm in a modularized way under the unified view described in

Section 2.1. Firstly, for the prompt design module that determines the expected output content, we borrow the proof completion module of RMaxTS to replace the proof-generation step of POETRY; then, for the response processing module, we perform proof structure analysis as in the proof augmentation procedure and try `<ATP>` for the failed **proof**...**qed** blocks or **by** clauses. A new node is added if the `<ATP>` method fails to solve the corresponding conjecture. For the state selection strategy, we follow the best-first strategy, but the priority is modified from the cumulative log-probability to the end position of the failed proof block, i.e., if one node is an ancestor of another, we prioritize the descendant; otherwise, we prioritize the one that appears first in the semi-proof.

In practice, to simplify the algorithm and suppress the sampling cost wasted in low-level details, we only try `<ATP>` and add new nodes for failed conjectures belonging to the original MCSP. Such simplifications make the above procedure become a plug-and-play ERP module over our ProofAug, as shown in Algorithm 2. This coincidence can be interpreted as that each conjecture naturally inherits one proof try from the original proof proposal. In other words, the ProofAug without ERP module is already implicitly a recursive proving method, in the sense that each compatible semi-proof exactly corresponds to one proof tree during the search process of POETRY if the two methods share the same final proof.

For an illustrative example of how the ERP module works in ProofAug, one can refer to Figure 1.

# 4. Experiments

A summary of our experimental results and comparison with previous methods is shown in Table 2. Below, we state the experimental setup and describe how different parts of the results are obtained and provide additional ablation study results. For the results in Lean implementation, please refer to Appendix C.

## 4.1. Benchmark

We evaluate our methods on the Isabelle part of miniF2F (Zheng et al., 2021), a benchmark for formal Olympiad-level mathematics problems across four different formal languages. Unless otherwise specified, we use 'miniF2F-test' or 'the original version of miniF2F-test' or 'the DSP-version miniF2F-test' to refer to the dataset included in the code repository[8] of Jiang et al. (2023), which provides an additional informal statement and informal draft

for each problem compared to the OpenAI miniF2F[9]. While both versions contain many mistakes and various extents of inaccuracies in the formal statements, previous work (Wang et al., 2023a; Zheng et al., 2024; Wang et al., 2024; Zhao et al., 2024) in general chooses to use this DSP-version for a fair comparison between different methods. Thus, in order to show the superiority of our ProofAug, we first follow previous work to do experiments on the DSP-version miniF2F-test.

However, as the performance of neural theorem provers has significantly improved recently, the ratio of unprovable theorems resulting from the mistakes to the total number of statements that provers fail to provide proofs becomes no longer ignorable, and is one of the possible reasons that Isabelle-based provers have fallen behind Lean-based ones on this cross-language benchmark. So, we build a curated dataset and part of our experiments are done on the curated version.[10]

## 4.2. Experimental Setup

The Isabelle version we use is Isabelle2022. We use the PISA environment (Jiang et al., 2021) to interact with Isabelle. Our `<ATP>` method is the combination of 8 Isabelle proof methods (*auto, simp, auto, blast, fastforce, eval, sos, arith, simp:field_simps, simp add:mod_simps*) and Sledgehammer. Following Jiang et al. (2023), we set the timeout for any proof step and Sledgehammer as 10s and 120s, respectively. During verification, we run 12 PISA instances in parallel on one Intel(R) Xeon(R) Gold 6326 CPU @ 2.90GHz[11].

The language model we use across all our experiments is deepseek-math-7b-base (Shao et al., 2024), run on Nvidia GeForce RTX 3090 with vLLM (Kwon et al., 2023). Unless otherwise specified, we follow Jiang et al. (2023) to use a sampling temperature $T = 0.6$ with top_p $= 0.95$. The default max number of tokens of the response is set to 2048, and when the length of the prompt exceeds 2048, we adjust the max number of tokens to $4096 - $#token_of_the_prompt.

For few-shot prompting (and similarly for zero-shot prompting), we prepare few-shot examples that are the same as those of Jiang et al. (2023) except that the proof sketches are modified to new full proofs written by us. For $n$-shots

---

[8] https://github.com/albertqjiang/draft_sketch_prove/blob/main/aligned_problems/new_complete_mini.jsonl

[9] https://github.com/openai/miniF2F/tree/main/isabelle

[10] You can find our curated version here: https://github.com/haoxiongliu/ProofAug/blob/main/isabelle_src/datasets/minif2f-test-curated.jsonl. See also Appendix H for the details of how we curate miniF2F.

[11] To reproduce our results, using the same CPU is a necessary requirement. See Appendix D for more discussion on the reproducibility issues.

Table 2: **Comparison of methods using Isabelle as the proof assistant on MiniF2F-test.** For BFS methods, the sample budget $N \times S \times T$ corresponds to $N$ attempts of $S$ expansion with $T$ iterations. As to tree-search methods, it becomes $N \times T$, with the same meanings for the symbols. Results with a '†' are obtained using a mixed strategy.

| Method | Model | Sample Budget | miniF2F-test |
|---|---|---|---|
| *Methods using Isabelle* | | | |
| DSP (Jiang et al., 2023) | CodeX | 100 | 39.3% |
| Subgoal-XL (Zhao et al., 2024) | Fine-tuned Llama-8B | 64 | 39.3% |
| | | 16384† | 56.1% |
| LEGO-Prover (Wang et al., 2023a) | mixed GPTs | 100 | 50.0% |
| Lyra (Zheng et al., 2024) | GPT-4 | 100 | 47.1% |
| | | 200 | 51.2% |
| POETRY (Wang et al., 2024) | Fine-tuned ProofGPT (1.3B) | $1 \times 32 \times 128$ | 42.2% |
| *Our Experiments (using Isabelle)* | | | |
| DSP baseline | deepseek-math-7b-base | 1 | 28.7% |
| | | 10 | 40.6% |
| | | 100 | 49.2% |
| ProofAug | deepseek-math-7b-base | 1 | 36.5%(+7.8%) |
| | | 10 | 44.7%(+4.1%) |
| | | 100 | 52.5%(+3.3%) |
| ProofAug (0-shot) | deepseek-math-7b-base | 500 | 54.5% |
| ProofAug (0-shot) + ERP | deepseek-math-7b-base | 500 | 56.1% |
| ProofAug + Mixed Strategy | deepseek-math-7b-base | 1400† | 61.9% |
| ProofAug + Mix. + Dataset Curation | deepseek-math-7b-base | 2100† | **66.0%** |
| *Methods using Lean* | | | |
| HTPS (Lample et al., 2022) | Evariste (600M) | $64 \times 5000$ | 41.0% |
| RMaxTS (Xin et al., 2024b) | DeepSeek-Prover-V1.5-RL (7B) | $32 \times 6400$† | 63.5% |
| BFS + CG (Wu et al., 2024) | InternLM2.5-StepProver (7B) | $256 \times 32 \times 600$ | 65.9% |

prompting, we randomly sample $n$ examples from the candidates. By default, we use 1-shot prompting. See Appendix E for our specific prompt text, the reason for choosing 1-shot, and more details on the prompt construction process.

For multiple attempts of proof, we record the checked semi-proofs and skip them when obtaining them in the subsequent attempts. This, in fact, harms the final performance since some tactics and Sledgehammer do not always give the same result under the same conditions. Nevertheless, this practice can help us significantly reduce the elapsed time of the experiments.

### 4.3. Results of ProofAug

We do ablation studies to validate the effectiveness of ProofAug (without the ERP module). We first get the DSP baseline results in our setting. Compared to the performance of the original DSP implementation on miniF2F-test, our DSP baseline is 9.9% higher (49.2% v.s 39.3% in 100 attempts) due to differences in proving environments[12], choice of the prompting methods and model selection. Then we compare ProofAug results with our DSP baselines. Under sample budgets of 1, 10 and 100, ProofAug achieves per-

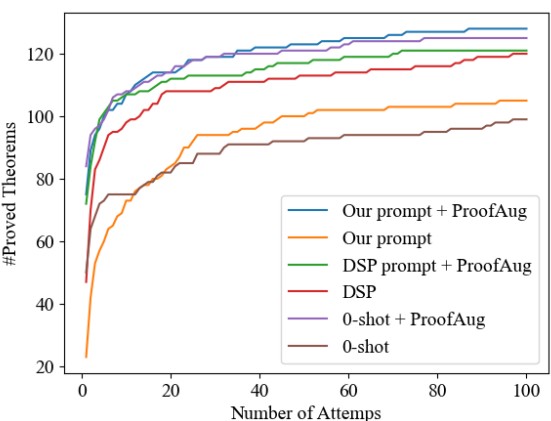

Figure 3: Ablation of three different prompting ways.

formance of 36.5%, 44.7% and 52.5%, significantly outperforming the baseline by 7.8%, 4.1% and 3.3%, respectively.

Note that ProofAug can also be applied to the proof sketches generated by DSP. Figure 3 shows how the number of proved theorems changes as the number of attempts increases w/ or w/o ProofAug for three prompting methods: DSP prompting, full proof prompting, and the more token-usage friendly zero-shot prompting. By w/o ProofAug,

---

[12]See Appendix D for a discussion on the differences of theorem proving environment between ours and that of Jiang et al. (2023).

we mean the 'Prove' step in DSP, i.e., `<ATP>` is tried at the **sorry**s in the proof sketch. It can be seen that while the performance of our few-shot prompting and zero-shot prompting w/o ProofAug is much lower than that of DSP prompting, ProofAug helps them achieve a comeback. This justifies our choice of generating full proofs instead of proof sketches.

### 4.4. Results of the ERP module

We try 500 attempts of proof for each problem, using zero-shot prompting for ProofAug w/ and w/o ERP module. When the ERP module is turned on, we record the cumulative number of queries and stop verifying when it reaches 500 for a fair comparison.

The result shows that the ERP module can bring an extra 1.6% performance gain. If we do not restrict the number of queries, we can achieve a pass rate of 57.4%(+2.9% compared to the ProofAug without ERP).

### 4.5. Cumulative Results of Mixed Strategy Experiments

During the completion of this work, we have tried various setups to search for the best recipe. While some setups fail to achieve the best performance on the benchmark compared to results we present above, diverse ways of generating proofs help prove more theorems. The setups include: 1) *Different versions of prompts.* We have three versions of demonstration examples for few-shot prompt generation, and two versions of zero-shot prompt are used in our experiments. 2) *Whether to include Informal drafts.* The informal drafts could mislead LLMs. Although ProofAug can sometimes be a rescue, it is not always the case.

In total, we try each problem of the original miniF2F-test under a sample budget of around 1400 in total, and 2100 for the two versions in total. See Table 3 in Appendix F for a breakdown of these two numbers. On the original miniF2F-test, we solve 61.9% of the problems, which is 5.8% higher than the previous Isabelle SOTA. On the curated version, the solving rate comes to 66.0%.

## 5. Discussion and Future Directions

Although this work demonstrates that our ProofAug method significantly improves the performance of the neural theorem prover on the miniF2F benchmark, there remains a notable gap in tackling IMO-level problems or research-level challenges within formal systems. How to incorporate ProofAug into expert iteration methods (Xin et al., 2024a) or online learning methods (Lample et al., 2022) to further improve the performance is one direction of future work.

We believe that another important direction for future work is to fully leverage the robustness introduced by ProofAug

through its ability to operate across various granularities and the fallback mechanism. Robustness is a highly desirable property for systems lacking inherent verification mechanisms, such as informal mathematical problem-solving and software engineering. Therefore, we are also highly interested in exploring the effective application of ProofAug in these domains.

## Impact Statement

This paper presents work whose goal is to advance the field of Machine Learning. There are many potential societal consequences of our work, none of which we feel must be specifically highlighted here.

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

# A. Other Related Work

In the main body, Table 1 and Table 2 only include representative neural theorem proving methods that are most related to this work: they all focus on solving Olympic-level mathematical problems with generative language models. Below we introduce other related work.

**Premise Selection.** One line of work uses embedding models to facilitate the selection of premises for automation tools (Irving et al., 2016; Wang et al., 2017; Goertzel et al., 2022; Yang et al., 2023). The machine-learning enhanced automation tools can be used to strengthen the power of the `<ATP>` method in our ProofAug algorithm. For example, the Magnushammer(Mikuła et al., 2024) is an ideal substitution for the sledgehammer, whose solving rate on miniF2F is 34.0% (against 20.9% of sledgehammer) and has a mode that does not require GPU or TPU accelerators. We believe that access to such tools can help ProofAug achieve better performance on theorem proving.

**Hierarchical Theorem Proving.** We note that the idea of hierarchically proving theorems in declarative ITP systems has been explored in several previous works. Besides the proof-step based recursive proving method POETRY (Wang et al., 2024) discussed in Section 3.3, LEGO-Prover (Wang et al., 2023a) uses a skill library to record the previously proved lemmas and retrieves relative skills when instructing GPT series models to synthesize its proof based on a decomposed informal draft. It is a hierarchical proving method in the sense that the lemmas used for proving the current theorem can be built on more low-level lemmas. From another point of view, LEGO-Prover uses all previous queries to solve the current theorem hierarchically. In contrast, Dong et al. (2024) does not record a skill library, but finishes the hierarchical proving from scratch. The key difference between these hierarchical proving methods and our ProofAug (without ERP module) is that we work in a bottom-up fashion to best exploit the power of traditional automation tools, contrary to their top-down frameworks that all require extra queries to the model when stepping into the next level proof. We argue that bottom-up fashion aligns more with the pre-trained corpus, thus making best use of the theorem proving knowledge possessed by the model. In contrast, top-down methods require in-context learning or fine-tuning to control the behavior of LMs, which could hurt the intrinsic ability of the model and bring extra complexity to the problem. This is likely the underlying reason why ProD-RL (Dong et al., 2024) fails to make significant improvements over the simple SFT method on miniF2F-test. Whereas our ProofAug serves as a play-and-plug module and can fall back to naive single-pass generation, being superior in efficiency, robustness, and flexibility. Moreover, by adding the ERP module, we integrate the top-down and bottom-up fashions in a sample-efficient way.

While finalizing this paper, the authors became aware of a concurrent and independent work Lu et al. (2024), which proposes PALM, that shares similar high-level ideas with ProofAug. However, our work is distinct in several key aspects: 1) PALM works in Coq (Barras et al., 1999), a procedural proof language where declarative modes (such as the 'Mathematical Proof Language' introduced in the Coq reference manual) are less common. Since tactics in the procedural style proof languages are designed in the spirit that the user decides what tactic to use next by observing the resulting goal or proof state after using the last tactic, it is often the case that new variables or facts are introduced by one tactic and the subsequent tactics need to specify the names of these facts. Thus, performing whole-proof generation as PALM does for Coq is generally harder than in the declarative style proofs and could be detrimental to the performance. Besides, in Coq proofs, there are few explicit intermediate conjectures or block structures as in those in Isar or Lean 4. Thus, for each tactic, the backtracking procedure of PALM simply goes back to the proof state before the unrepairable error and continues backtracking, except for the bullet point structure that is equivalent to the 'case' structure in Isabelle/Isar (rather than the more common 'have' method). Consequently, the proofs found by PALM mostly show scarce high-level structures as we can observe from the results provided in its code repository [13]. In contrast, our ProofAug departs from the DSP framework and takes an 'augmentation' perspective to solve the issues of DSP. We first determine all compatible semi-proofs, then sequentially prune the semi-proofs that become impossible to be extended into a full proof after each time we apply `<ATP >` to an intermediate conjecture. This procedure makes ProofAug fit with the 'sketch' procedure of DSP that translates the informal proof into intermediate conjectures, and each semi-proof is expected to align with the corresponding part of the informal proof. Note that such correspondence is hard to achieve for procedural proofs and obligating the LLMs to use the uncommon declarative mode or 'assert' keywords in Coq could be detrimental to the performance. Although in Appendix C we also discuss how to transform a procedural style proof language into a declarative one in the spirit described in Wiedijk (2012) and then train models and apply ProofAug accordingly, this procedure needs to fine-tune a model on enough data, which is in general expensive. In total, ProofAug is the choice for theorem proving models such as the DeepSeek-Prover series and the Kimina-Prover (Wang et al., 2025) series that achieve SOTA performance on the cross-language benchmark miniF2F, and provide detailed solutions

---

[13] https://github.com/lachinygair/PALM/tree/main/evaluation/proof

and code. 2) Our work further explores the Efficient Recursive Proving module and shows it can boost the performance with a relatively small number of queries. In fact, the counterpart of the ERP module for procedural-style proofs should be some kind of truncate-and-resume tree-search algorithm (that only adds new nodes based on the initial proof proposal) proposed in Xin et al. (2024b). However, since the proof style adopted by Deepseek-Prover-V1.5 series models is rather declarative, RMaxTS becomes inefficient for them due to the potential reasons we analyzed in Section 3.3. 3) We make extra contributions such as exploring extended mixture of strategies to enhance the performance, curating the Isabelle version of miniF2F-test and finally achieving a cumulative pass rate of 66.0% on this benchmark, while the performance of Coq-based methods remains unclear on the problems of miniF2F since the Coq version is not included by now.

**Declarative versus Procedural.** The discussion of proof styles dates back to Harrison (1996). Wiedijk (2006) compares the proof styles of a wide range of different systems. Wiedijk (2012) discusses how to convert from a procedural system to a declarative one. Avigad et al. (2021) introduces the two styles of proof in Lean and provides concrete examples.

**Other proof-step generation works.** Aside from works introduced in Table 1, such as Polu & Sutskever (2020); Lample et al. (2022), Han et al. (2022); Polu et al. (2022); Wang et al. (2023b) are also valuable explorations on what practice is beneficial for tree-search proof-step generation methods. Thakur et al. (2024) designs an agent that makes use of the history ITP feedbacks in a serialization way.

**Other whole-proof generation works.** Xin et al. (2024a); Wu et al. (2022) verify the effectiveness of expert iteration on neural theorem proving using the naive whole-proof generation method. They build the first-round theorem statements through autoformalization of the informal problems. First et al. (2023) employs Isabelle error messages to enhance the direct full proof synthesis ability of LLM.

# B. Examples of the DSP Issues

Figure 4 and Figure 5 show examples of the 'Hard Conjecture' issue and the 'Complicated Draft' issue described in Section 1.

```
theorem numbertheory_sqmod3in01d:
  fixes a :: int
  shows "a^2 mod 3 = 0 \<or> a^2 mod 3 = 1"
proof -
(* Let a be an integer, then a \pmod 3 \in {0, 1, 2}. *)
  have c0: "a mod 3 \<in> {0,1,2}" by fastforce
(* Using that a^2 \pmod 3 = (a \pmod 3)^2 \pmod 3 *)
  have "a^2 mod 3 = (a mod 3)^2 mod 3" by (simp add: power_mod)
(* we have a^2 \pmod 3 \in {0, 1}. *)
  then show ?thesis using c0 sledgehammer
qed

Isabelle Info: Sledgehammering... No proof found
```

Figure 4: **An example of the 'Hard Conjectures' issue.** In this example, although the informal draft is reasonable to human, sledgehammer fails to find a proof for the final step.

# C. Discussion of Implementing ProofAug for Other Proof Assistants

**Lean.** Although Lean proofs can be completely tactic-based, block-structures starting with *by* are very common as well, especially for the proofs of Olympics-level math problems (refer to Xin et al. (2024b); Wang et al. (2025) for the example proofs they find). As a result, Lean proofs are also amendable for proof structure analysis, so we also build a simple implementation of ProofAug in Lean. As to the substitution of the `<ATP>` method, we use the combination of Aesop(Limperg & From, 2023), omega and our hand-made heuristic `try norm_num [*]; try field_simp [*] at *; try ring_nf at *; try nlinarith`. For the proof structure analysis process, given a line of code, we manually infer whether it is intended to be part of a statement or tactic from its indent and context. Our experiment result shows that ProofAug improves the performance of Kimina-Prover-Preview-Distill-1.5B from 44.3% to 50.4% on miniF2F. The improvement can be further enhanced if we can obtain a more proper control of the parsing and structure analysis

```
    theorem
      "gcd 180 168 = (12::nat)"
      by eval
- - - - - - - - - - - - - - - - - - - - - - - - - - - - - - - - - - - - - - -
    theorem
      "gcd 180 168 = (12::nat)"
    proof -
  (* If a number divides into both 180 and 168 *)
      have "gcd 180 168 dvd 180" sledgehammer
      moreover have "gcd 180 168 dvd 168" sledgehammer
  (* it must also divide into their difference. *)
      finally have "gcd 180 168 dvd 12" sledgehammer
    qed

    Isabelle Info: Sledgehammering... No proof found
```

Figure 5: **An example of the 'Complicated Draft' issue.** In this example, while the `eval` proof method can prove the theorem easily in Isabelle, the translation of informal draft into a formal proof fails to pass the verification due to the lack of explicit type annotations, which is a typical error the language models could make in autoformalization.

process and get better substitutions of the `<ATP>` method. Readers may refer to our code for more implementation details.

For procedural ITPs that work mostly in a tactic-based way and do not have block structures in the proofs explicitly, in principle, one can follow Wiedijk (2012) to build a proof interface that supports declarative style proofs upon the original system. Since Wiedijk (2012) also provides a generic way to convert the proofs in the original procedural language to declarative ones, one may obtain data for training language models on this new language by conversion of the corpora in the original language. Then one can perform ProofAug on this new declarative language.

## D. Theorem Proving Environment Setup Details and Discussion on the Reproducibility

Our theorem proving environment setup mainly follows that described in Jiang et al. (2023), but there are some subtle differences. For the setup of PISA, we set the candidate provers of Sledgehammer to `CVC4 vampire verit e spass z3 zipperposition`, with a 30s timeout for each and 120s in total, as in Jiang et al. (2023). However, we find that in their PISA source code, they use cvc5 instead of CVC4 and the actual total timeout for Sledgehammer is around 35s instead of the intended 120s, seemingly due to an implementation bug. We fix this bug and still use CVC4 since cvc5 is not integrated in Isabelle 2022. We do not compile the Archive of Formal (AoF) library due to lack of disk space to simultaneously run 12 instances possessing compiled AoF heaps on our server. As a result, we do not import the `Symmetric_Polynomials.Vieta` theory as in Jiang et al. (2023). Besides, The heuristics used in our `<ATP>` method and those in Jiang et al. (2023) differ in that we add an extra `simp:add mod_simps`, remove `force` since we already have `fastforce` and remove `presburger,linarith` since `arith` integrates the two in a rough sense.

To reproduce the results that involve the use of complex built-in proof methods or off-the-shelf ATPs, one has to make sure to run PISA instances on the same CPU since the performance of these automation methods heavily rely on the computational power of the CPU.

Nevertheless, even using the same environment, it is still hard to produce exactly the same results due to the inherent randomness of these automation methods. The good news, however, is that for theorem proving, we are confident that we have made some progress when new theorems get proved, no matter whether this is completely reproducible or not. So we focus on making sure the obtained proofs are correct by verifying them again in Isabelle. We provide the found proofs in attachment to our code. Note that although there are cases where the reconstructed proof fails due to the timeout of Isabelle built-in methods such as `smt` and `metis`, it does not mean that the theorems are not proved. It is only the limitation of these tactics.

# E. Prompt Construction

The prompts in our experiments are all constructed from three components: prompt template, few-shot demonstration examples. and a prompter. The prompt template is a JSON file that contains different roles as names and corresponding templates as values. Each example is also an JSON object whose names occur in the templates. A prompter mainly consists of a chat template that maps the sampled examples to a conversation consisted by **messages** according to the prompt template.

For example, for few-shot prompting for ProofAug, we use the prompt template shown in Figure 6. The examples contain names `xi`, `yi`, `xf`, `yf`, corresponding to an informal statement, an informal draft, the formal statement, and the formal full proof. For zero-shot prompting (possibly with ERP), we use the prompt template in Figure 8, which has two extra fields `ps` and `yf_p`. For non-ERP prompting, they are set to null string. For ERP prompting at state $s$, they corresponds to the 'proof state' $s.state$ and the partial proof that have been verified at this state.

As to the chat template, we simply use three newlines to separate pairs of user-assistant messages and concatenate the user message and assistant message in each pair. Nevertheless, we mainly expect the response to stop at the end of the Isabelle code block.

Figure 7 shows an example of the sampled 1-shot prompt.

```
{
    "user": "Please complete the following Isabelle proof.\n```isabelle\n(*
        Informal statement:\n{xi}\n\nInformal proof:\n{yi} *)\n{xf}\n",
    "assistant": "{yf}\n```\n\n\n",
    "stop": ["```", "\ntheorem", "\nend"]
}
```

Figure 6: The prompt template we use for few-shot prompting.

As to the choice of the number of shots to use, in Jiang et al. (2023), they set #shot=3 when prompting the model to translate a draft. We find that since the maximum sequence length of deepseek-math-7b-base is only 4096, few-shot prompting can lead to failure of proofs that are long in nature. As a consequence, DSP with 3-shots prompting solves 117/244, 2-shots 118/244, both smaller than 1-shot (120/244). Thus, across the paper, we use 1-shot prompting unless otherwise stated.

# F. Breakdown of the Mixture of Strategies

Table 3 shows the breakdown of the mixture of strategies in Section 4.4. Below are some explanations of the options appearing in Table 3.

As to the prompt version, we have 3 versions of few-shot prompting and two versions of zero-shot prompting. The three versions of few-shot prompting share the same prompt template shown in Figure 6, but differ in examples: 'DSP' refers to the examples in Jiang et al. (2023), and 'few-shot' refers to examples used for ProofAug, where we write full proofs that strictly match the informal proofs provided in the miniF2F dataset of Jiang et al. (2023). In contrast, many of the informal proofs of DSP few-shot prompting examples are just descriptions of formal sketches prepared in advance. We also have a 'few-shot-legacy' version where we first write the simplest proof (rather than a proof sketch) we can come up for each theorem, then write a natural language description for it as an informal proof. Although it turns out that the 'few-shot' version is the best one for 1-shot prompting, the three sets of examples form a complement to each other. Since zero-shot prompting do not involve examples, the two versions of our zero-shot prompts only differ in the instruction text and format. For the 0-shot-legacy prompt, we do not have the 'Below is a complete proof of a statement in Isabelle/Isar, translated from the informal proof.' text in the "user" field shown in Figure 8.

A '×' in the 'Draft' column means that informal statement and the informal proof are not included in the prompt.

For the experiments using the ERP module, we only count the proved theorem using fewer than 700 and 200 requests to the model, for the experiments with 500 attempts and 100 attempts, respectively. As to the pass rates of these experiments, since we only evaluate on the previously unproved theorems for the curated version dataset, we put N/As for them.

For further information, such as the contents of the examples, please refer to our code.

```
Please complete the following Isabelle proof.
(* Informal statement:
When Rachel divides her favorite number by 7, she gets a remainder of 5.
What will the remainder be if she multiplies her favorite number by 5 and then divides by 7?
Show that it is 4.

Informal proof:
Let n be Rachel's favorite number. Then n ≡ 5 (mod 7), so 5n ≡ 5 · 5 ≡ 25 ≡ 4 (mod 7). *)
theorem mathd_numbertheory_335:
  fixes n :: nat
  assumes h0 : "n mod 7 = 5"
  shows "(5 * n) mod 7 = 4"
proof -
  (* so 5n ≡ 5 · 5 ≡ 25 ≡ 4 (mod 7). *)
  have c0: "(5 * n) mod 7 = (5 * 5) mod 7"
  proof -
    have c00: "(5 * n) mod 7 = (5 mod 7 * (n mod 7)) mod 7"
      by (auto simp: mod_simps)
    then show ?thesis unfolding h0
      by auto
  qed
  also have "... = 4 mod 7" by auto
  finally show ?thesis by auto
qed

Please complete the following Isabelle proof.
(* Informal statement:
{informal statement of the problem}

Informal proof:
{informal draft of the problem} *)
{formal statement of the problem}
```

Figure 7: An example of final 1-shot prompt for ProofAug.

# G. Additional Experimental Results

On miniF2F-valid, ProofAug w/ and w/o ERP module achieve 58.6% and 57.0% in 100 attempts of proof respectively. During the experiment, we make sure the problem to prove is not in the candidate examples.

# H. Curation of the Isabelle part of miniF2F

Our curation of DSP-version miniF2F test split mainly includes two types: 1) Typo fix: Statements with typos in variable names or numbers are mostly completely wrong and unprovable. 2) Modifying the type of the variables from `nat` to `int` for theorems that could potentially involve the minus operation in the proofs. This is important because, for example, the operation `(2::nat) - (3::nat)` is valid and expected to be `(0::nat)` in Isabelle/HOL. Although sometimes the original statement can still be proved in the sense that the resulting 0 also satisfies the constraints in the statement, it does not express the intent of the original problem. Figure 9 shows such an example.

After finishing the curation process, the authors get to learn about that there is a FAIR version repository of miniF2F[14] that is actively in maintenance and most of the typos and inappropriate uses of minus of `nat` type numbers we find in the DSP-version have been fixed in the latest commits. Nevertheless, several typos and some inappropriate use of `nat` type remain. Thus, we still make a pull request[15] to the FAIR miniF2F repository to update part of our curation to the upstream.

---

[14]https://github.com/facebookresearch/miniF2F
[15]https://github.com/facebookresearch/miniF2F/pull/18

```
{
    "user": "Please complete the following Isabelle proof.\n```isabelle\n(*
        Informal statement:\n{xi}\n\nInformal proof:\n{yi} *)\n{xf}\n{yf_p}{ps}",
    "assistant": "{yf_c}\n```\n\n",
    "stop": ["```", "\ntheorem", "\nend", "\nlemma"]
}
```

Figure 8: The prompt template we use for zero-shot prompting for efficient recursive proving (ERP) module.

Table 3: Breakdown of the mixture of strategies introduced in Section 4.4.

| #attempts | Prompt Ver. | #shots | Draft | ERP | #queries | Curated Ver. | Pass Rate |
|---|---|---|---|---|---|---|---|
| 500 | 0-shot | 0 | × | × | 500 | ✓ | N/A |
| 500 | 0-shot | 0 | ✓ | ✓ | 700 | × | 57.4% |
| 100 | few-shot | 1 | ✓ | × | 100 | × | 52.5% |
| 100 | few-shot | 1 | ✓ | × | 100 | ✓ | N/A |
| 100 | few-shot-legacy | 1 | × | × | 100 | × | 48.4% |
| 100 | few-shot-legacy | 1 | ✓ | × | 100 | × | 50.8% |
| 100 | DSP | 1 | ✓ | × | 100 | × | 49.2% |
| 100 | 0-shot-legacy | 0 | ✓ | × | 100 | ✓ | N/A |
| 100 | 0-shot-legacy | 0 | ✓ | × | 100 | × | 50.8% |
| 100 | 0-shot-legacy | 0 | ✓ | ✓ | 200 | × | 52.0% |

# I. An Example of How ProofAug Induce a Proof

Figure 10 shows an example of ProofAug inducing a proof from a failed proof proposal generated by the model.

```
Differences:
 theorem imo_2001_p6:
-  fixes a b c d ::nat
+  fixes a b c d ::int
   assumes "0 < a \<and> 0 < b \<and> 0 < c \<and> 0 < d"
     and "d < c"
     and "c < b"
     and "b < a"
-    and "a * c + b * d = (b + d + a - c) * (b + d - a + c)"
+    and "a * c + b * d = (a + b - c + d) * (- a + b + c + d)"
   shows "\<not> prime (a * b + c * d)"
```

Figure 9: **How we fix the imo_2001_p6 in DSP-version miniF2F-test.** The "a * c + b * d = (b + d + a - c) * (b + d - a + c)" constraint does not express what it intends in the original statement when $a > b + d$.

```
theorem algebra_sqineq_at2malt1_init_proof:
  fixes a::real
  shows "a * (2 - a) \<le> 1"
proof -
  have "(a - 1)\<^sup>2 \<ge> 0" for a::real
  proof -
    have "0 \<le> (a - 1) * (a - 1)"
      using zero_le_square by auto
    then show "(a - 1)\<^sup>2 \<ge> 0"
      by (simp add: power2_eq_square)
  qed
  then have "a * (2 - a) \<le> 1" for a::real
  proof -
    have "a * (2 - a) = 2 * a - a\<^sup>2" by (simp add: power2_eq_square)
    also have "... = (a - 1)\<^sup>2 + 1 - a\<^sup>2" by (simp add: algebra_simps)
    also have "... \<le> 1"
      using \<open>0 \<le> (a - 1)\<^sup>2\<close> by linarith
    finally show ?thesis .
  qed
  then show ?thesis .
qed
```

```
theorem algebra_sqineq_at2malt1_MCSP:
  fixes a::real
  shows "a * (2 - a) \<le> 1"
proof -
  have "(a - 1)\<^sup>2 \<ge> 0" for a::real
  proof -
    have "0 \<le> (a - 1) * (a - 1)"
      using zero_le_square by auto
    then show "(a - 1)\<^sup>2 \<ge> 0"
      by (simp add: power2_eq_square)
  qed
  then have "a * (2 - a) \<le> 1" for a::real
  proof -
    have "a * (2 - a) = 2 * a - a\<^sup>2" sorry
    also have "... = (a - 1)\<^sup>2 + 1 - a\<^sup>2" sorry
    also have "... \<le> 1"
      using \<open>0 \<le> (a - 1)\<^sup>2\<close> sorry
    finally show ?thesis .
  qed
  then show ?thesis .
qed
```

```
theorem algebra_sqineq_at2malt1_final:
  fixes a::real
  shows "a * (2 - a) \<le> 1"
proof -
  have "(a - 1)\<^sup>2 \<ge> 0" for a::real
  proof -
    have "0 \<le> (a - 1) * (a - 1)"
      using zero_le_square  by auto
    then show "(a - 1)\<^sup>2 \<ge> 0"
      by (simp add: power2_eq_square)
  qed
  then have "a * (2 - a) \<le>  1" for a::real by sos
  then show ?thesis .
qed
```

Figure 10: **An example of ProofAug inducing a proof from a failed initial proof proposal.** It can be seen that the initial proof fails inside the **proof**...**qed** block of proving $a(2 - a) \le 1$ due to the wrong intermediate claim $2 * a - a^2 = (a - 1)^2 + 1 - a^2$. ProofAug finds that $a(2 - a) \le 1$ can be directly proved with the proof method `sos` given the previous proved fact $(a - 1)^2 \ge 0$.

