# OpenReview forum: "ProofAug: Efficient Neural Theorem Proving via Fine-grained Proof Structure Analysis"
_ICML.cc/2025/Conference — ICML 2025 poster_

### Official Review · Reviewer_6tf8 · 2025-03-13

**Overall Recommendation:** 3

**Summary:**

The paper introduces ProofAug, a novel approach for enhancing neural theorem proving (NTP) by integrating large language models (LLMs) with traditional automation tools. Unlike prior approaches such as the Draft, Sketch, and Prove (DSP) framework, which generate rough proof sketches and rely on automation tools to fill gaps at a single granularity, the paper proposes starting with a full proof generated by an LLM and employs a fine-grained analysis to derive a maximal compatible semi-proof (MCSP)—a structurally intact proof verifiable by an interactive theorem prover (ITP) like Isabelle, with some unproven parts marked as "sorry." These gaps are iteratively refined using automation tools at varying levels of detail, with the process repeating recursively until a complete proof is achieved or deemed infeasible. The method is enhanced by an efficient recursive proving (ERP) module, which integrates with tree-search algorithms to boost sample efficiency, achieving a notable 66.0% pass rate on the miniF2F-test benchmark after dataset curation (up from 61.9% originally), surpassing previous state-of-the-art results of 50-60%.

**Claims And Evidence:**

C1: Limitations in DSP where generated proofs may produce overly difficult intermediate conjectures or overly complex formal proofs

The authors discuss limitations of the DSP framework in the introduction and as motivation for their approach. However, aside from a couple of examples, there doesn't seem to be any quantitative evidence of these being primary limitations. To be clear, intuitively these do make sense but there doesn't seem to be any quantitative evidence.

C2: Novel approach ProofAug which improves sample efficiency and can be seamlessly integrated with tree search algorithms.

The method discussed in Section 3, is evaluated on miniF2F and achieves better performance than DSP given the same budget. Additionally, the results with the ERP module also demonstrate strong performance.

C3: State-of-the-art performance with a mixed strategy

I find this claim a bit problematic. The authors acknowledge that this number is on a curated version of the data, however, the baselines numbers as far as I can tell are not computed on this curated data. This makes the comparison unfair and consequently the claim of SoTA seems unsubstantiated.

**Essential References Not Discussed:**

N/A

**Experimental Designs Or Analyses:**

As mentioned above, I believe the results on the curated data with a mixture strategy are compared unfairly as the numbers for the proposed method are computed on a different (curated) version of the data. Moreover, it is not clear what the curation procedure is exactly but I appreciate the authors including the curated dataset with the submission. Additionally, the experiments are limited to Isabelle, and though the authors talk about other systems in the Appendix, some concrete experiments would be helpful.

**Methods And Evaluation Criteria:**

I am not an expert on the topic but miniF2F is a standard dataset used for evaluating neural theorem provers.

**Other Comments Or Suggestions:**

N/A

**Other Strengths And Weaknesses:**

* (Strength) The unified perspective on theorem proving with language models in Section 2.1 was quite interesting and insightful.
* (Strength) The authors include code to reproduce the experiments.

**Questions For Authors:**

Could you please clarify the choices for curation, and what is the rationale behind the comparison to other methods on a different dataset?

**Relation To Broader Scientific Literature:**

The paper improves upon a widely used approach of DSP for neural theorem proving and provides a new interesting approach for combining LLMs with existing automated theorem provers.

**Theoretical Claims:**

N/A

---

> ### Author Rebuttal · Authors · 2025-03-31
>
> We thank the reviewer for the constructive feedback. Below is our response to your concerns:
>
> **Q1**: The authors discuss limitations of the DSP framework in the introduction and as motivation for their approach. However, aside from a couple of examples, there doesn't seem to be any quantitative evidence of these being primary limitations. To be clear, intuitively these do make sense but there doesn't seem to be any quantitative evidence.
>
> **A1**: Thank you for raising the concerns about quantitative evidence. This is a nuanced issue here. Firstly, we would like to clarify that the two limitations we discuss in our paper are not meant to classify the error reasons of the failed verification in DSP and claim which ones are primary. Instead, as stated in line 59, our argument is that "it(DSP) may also result in proof failures in unintended ways". By 'unintended', we mean the situations where the there is a chance for the direct whole-proof completion counterpart to generate a correct proof, while the DSP method leads to the failure.
> In fact, any syntactically correct DSP sketch that fails to pass the verification must fall into either or both of the following two categories: 1. some provable intermediate conjecture cannot be solved by <ATP>. 2. some intermediate conjecture (which is intended to be the translation of a step in the informal draft in DSP) is unprovable in Isabelle/HOL due to being incorrect or being improperly expressed in Isabelle/HOL. The first situation corresponds to our 'Hard Conjecture' issue, and direct whole-proof completion always has a chance to solve the conjecture. As to the second one, for direct whole-proof completion, there is also always a chance that if we do not follow the informal draft to propose this intermediate conjecture, we will get rescued.  This is the 'Complicated Draft' issue. As a result, the two issues we propose can almost cover 100\% of the 'unintended' failures of DSP.
> We apologize for the confusion caused by the lack of clarification of what 'unintended' means in our paper and will add more details to the revised version.
>
> If you are interested in how many such intended failures can be rescued by ProofAug, i.e. cases where the chances are actually realized in practice (the function of <ATP> in ProofAug can be seen as to help find the alternative proof), Figrue 3 can be a reference (DSP v.s 0-shot/Ours+ProofAug for different number of attempts). Remember that it does not reflect the ratio of different types of failures in DSP.
>
>
> **Q2**：(State-of-the-art performance with a mixed strategy). The authors acknowledge that this number is on a curated version of the data, however, the baselines numbers as far as I can tell are not computed on this curated data. This makes the comparison unfair and consequently the claim of SoTA seems unsubstantiated. Moreover, it is not clear what the curation procedure is exactly. Could you please clarify the choices for curation, and what is the rationale behind the comparison to other methods on a different dataset?
>
> **A2**: In Table 2, we have also reported our results on the original Isabelle version of minif2f (61.9\%) and show that our method is superior to other methods using Isabelle (previous Isabelle SOTA on minif2f-test is 56.1\%). The goal of curating the dataset is for the comparison with results evaluated on the Lean versions of minif2f. Our curation mainly consists of two types: 1. Fix typos. 2. Modify the type of the variables from Nat to Int for theorems that perform division on the variables (refer to our response to Q2 of Reviewer rCPC for detailed explanation of this type of curation). We have manually checked a large part of the Lean version of minif2f and find that no issues falling into these two types appear. Thus, we believe that using the evaluation result on our curated Isabelle version of minif2f to compare with the Lean version result is more fair.
>
>
> **Q3**: The experiments are limited to Isabelle, and though the authors talk about other systems in the Appendix, some concrete experiments would be helpful.
>
> **A3**: We are currently conducting our plans to implement and evaluate ProofAug on Lean. Preliminary results show that using heuristic tactic combinations as the <ATP> to build a Lean version of ProofAug can help improve about 3\% on the pass@1 rate for Deepseek-Prover-V1.5-SFT, but hard to make improvements for pass@128 or higher number of attempts. Integration with hammer tools in Lean, such as lean-smt, is necessary for further improvements. Also, some additional modifications might be needed (You can refer to our response to Q5 of Reviewer vgEx for more details). In the revised version, we will update some progress and results we finish by then.
>
>
> Again, we thank the reviewer for the valuable comments. We hope that our responses have addressed your concerns.

---

> > ### Comment · Reviewer_6tf8 · 2025-04-02
> >
> > Thanks for the clarifications! I maintain my positive assessment of the work!

---

### Official Review · Reviewer_vgEx · 2025-03-13

**Overall Recommendation:** 4

**Summary:**

Recursive theorem decomposition and rebuilding to obtain SOTA scores on miniF2F-test

## update after rebuttal
Satisfied with responses to questions : Rating remains "4: Accept"

**Claims And Evidence:**

* Claim: ProofAug enjoys superior sample efficiency.
  + Table 2 shows that ProofAug outperforms the DSP baseline at sample budgets of 1, 10, and 100 queries on miniF2F-test.
* Claim: ProofAug achieves a new SOTA on miniF2F-test.
  + Table 2 compares ProofAug's performance to previous SOTA methods on miniF2F-test, though this 66.0% figure appears to have been achieved through a 'mixed strategy', i.e. throwing a variety of techniques at the solution (explained in Appendix F), rather than 'pure' ProofAug.
* Claim: ProofAug is a versatile plug-and-play module and integrates well with tree-search algorithms, exemplified by the ERP module.
  + The paper describes ProofAug as a module within a unified view of theorem proving (Section 2.1) and demonstrates its integration with an ERP module (Section 3.3, Algorithm 2). Experimental results though (surprisingly) make it seems like it produces only a marginal gain.
* Claim: ProofAug addresses the "Hard Conjectures" and "Complicated Draft" issues.
  + Starting from full proof generation aims to mitigate "Hard Conjectures" and the recursive coarsening of semi-proofs addresses "Complicated Draft", this important aspect is left to two Figures in Appendix B.

**Essential References Not Discussed:**

References seem comprehensive

**Experimental Designs Or Analyses:**

* Control of Variables: The paper mentions using the same `deepseek-math-7b-base` model, Isabelle version, and PISA environment across experiments, which ensures fair comparisons. The discussion in Appendix D regarding reproducibility and CPU dependency is also important for experimental validity.
* Query Limits and Sample Budgets: The paper clearly reports sample budgets and query limits, allowing for a fair assessment of sample efficiency.
* Ablating ProofAug from different prompting strategies (DSP, full proof, zero-shot) effectively demonstrated the contribution of ProofAug across various prompting approaches.

**Methods And Evaluation Criteria:**

Benchmarks/evaluations make sense - and obtaining good miniF2F-test scores coupled with an eye to efficiency is good.

**Other Comments Or Suggestions:**

* Abstract: "with a total sample budget of only 2100" :: No sense of units here : Is this cumulative number of calls to the LLM?  Over how many questions?  (Abstract should stand alone)
* Figure 3 label : "Number of Attemps" -> "Number of Attempts"
* Section 4.1: "For multiple attempts of proof," -> "For multiple attempts of a given proof," (my understanding here)
* Section 4.1: "do not always give the same result under the same condition" -> "do not always give identical results under the same conditions"
* Section 4.4: "During the completion of this work, we have tried various" -> "During the completion of this work, we tried various"

**Other Strengths And Weaknesses:**

Strengths:

* Originality: The core idea of fine-grained proof structure analysis and MCSP for enhancing neural theorem proving is novel and offers a distinct approach compared to existing methods.
* Significance: The significant improvement in sample efficiency and achieving good results on miniF2F-test demonstrates the practical impact and potential of ProofAug.
* Comprehensive Evaluation: The paper provides a thorough experimental evaluation with ablation studies.  The overall claims are supported.

Weaknesses

* SOTA Result: While the overall claims are supported, the headline SOTA result (since it is based on a 'Mixed Strategy' rather than solely the proposed technique in isolation) somewhat *weakens* the overall idea : The recursive decomposition is interesting enough...

**Questions For Authors:**

* Section 4.4 : "Besides, we find some incorrect theorem statements in the miniF2F-test dataset during our experiments, so we build a curated dataset and part of the experiments are done on the curated version." : Have these been reported 'upstream'?
* While Appendix C discusses the generic applicability, are there concrete plans to implement and evaluate ProofAug on other proof assistants beyond Isabelle, such as Lean? If so, what are the main challenges anticipated in porting ProofAug to a different ITP environment, and how might the performance differ?

**Relation To Broader Scientific Literature:**

* Builds upon DSP Framework (Jiang et al., 2023): ProofAug directly addresses limitations observed in the Draft, Sketch, and Prove (DSP) framework
* Addresses Sample Efficiency in Whole-Proof Generation: The paper acknowledges the sample inefficiency in whole-proof generation methods and positions ProofAug as a solution to improve. This relates to the broader challenge of making LLM-based theorem provers more practical
* Integrates Automation Tools (ATPs): ProofAug strategically integrates off-the-shelf ATPs
* Related to Tree-Search in Proof Proving (AlphaGo, etc): The paper explicitly connects ProofAug to tree-search algorithms and demonstrates its compatibility with such approaches through the ERP module

**Theoretical Claims:**

The paper is primarily empirically driven and does not make significant theoretical claims that require formal proof verification.

---

> ### Author Rebuttal · Authors · 2025-03-31
>
> We appreciate the valuable comments from the reviewer, and especially thank you for recognizing and expressing interest in the experimental details in our code. Below is our response to your concerns:
>
> **Q1**: Experimental results of the ERP module make it seem like it produces only a marginal gain.
>
> **A1**: We agree with you that the 1.6\% improvement seems like only a marginal gain. However, a 1.6\% improvement under a sample budget (number of calls to LLM for each problem) of 500 is already a significant improvement when compared with RMaxTS, which only improves around 0.1\% for Deepseek-Prover-V1.5-SFT under a sample budget of 3200 (Even when the sample budget comes to 16×6400, the improvement is only 1.6\% as well). It is the overall difficulty of using tree-search methods to outperform simple resampling under small sample budgets that make you surprised at the seemingly marginal gain.
>
>
> **Q2**: While the overall claims are supported, the headline SOTA result (since it is based on a 'Mixed Strategy' rather than solely the proposed technique in isolation) somewhat weakens the overall idea.
>
> **A2**: We agree that the overall idea will be strengthened if the SOTA result is reported without using mixed strategies. For 'Pure' ProofAug (with ERP module), we have also achieved SOTA within isabelle-based methods (note that the previous SOTA Isabelle-based method, SubgoalXL, also uses 'mixed strategies' that even include using multiple trained checkpoints), which justifies the superiority of the proposed technique in isolation. We will highlight this in the revised version.
>
>
> **Q3**: (Suggestions on the writing and some specific expressions)
>
> **A3**: Thank you for the suggestions. We will modify "with a total sample budget of only 2100" to "restricting the number of LLM queries to 2100 for each problem" in the abstract and fix other typos and grammar issues in the revised version.
>
>
> **Q4**: Section 4.4 : "Besides, we find some incorrect theorem statements in the miniF2F-test dataset during our experiments, so we build a curated dataset and part of the experiments are done on the curated version." : Have these been reported 'upstream'?
>
> **A4**: Thank you for the suggestion, and we agree that it is important to report them upstream. However, since there is a gap between the verification style of DSP (which always imports a same set of theories for every theorem) and that of the original minif2f dataset (where different theories are imported for each theorem), extra efforts are needed to make a pull request to the minif2f repository. We will report our curation upstream as we finish the transfer to the original minif2f dataset style.
>
>
> **Q5**: While Appendix C discusses the generic applicability, are there concrete plans to implement and evaluate ProofAug on other proof assistants beyond Isabelle, such as Lean? If so, what are the main challenges anticipated in porting ProofAug to a different ITP environment, and how might the performance differ?
>
> **A5**: Yes, we are currently conducting our plans to implement and evaluate ProofAug on Lean. The main challenges we are facing are the engineering complexities involved in the implementation. To name a few:
> 1. The hammer tools (such as the lean-smt project) for lean are still in beta. So it takes time to make the whole pipeline work.
> 2. We need to adapt many implementation details from Isabelle/HOL to Lean. For example, there are no concepts like 'proof mode' in Lean so our Lean version implementation could no longer rely on them.
>
> We anticipate that the performance improvement might be less than that in Isabelle to some extent, since the current Lean hammers are weaker, and Lean proofs are usually in the mixed declarative and procedural style rather than being mostly declarative like Isabelle/Isar proofs. Modifications to ProoAug are probably needed to make it behave well for Lean if we do not choose to derive a completely declarative proof language from Lean as described in Appendix C.
>
> As a result, we feel like ProofAug should better represent the overall idea of 1. get a list of semi-proofs from the proof proposal 2. apply hammers for the semi-proofs in an organized way 3. an optional recursive proving module, while the specific implementations differ for different proof systems. If you think this statement can help people understand the idea of ProofAug better, we can add it to the revised version, and refer to Algorithm 2 as the "Implementation of ProofAug in Isabelle/HOL".
>
>
> Finally, we would like to thank you for your time and consideration. We hope that our responses have addressed your concerns.

---

> > ### Comment · Reviewer_vgEx · 2025-04-04
> >
> > Thanks for the answers to my questions.  My rating remains "4: Accept"

---

### Official Review · Reviewer_u8sL · 2025-03-13

**Overall Recommendation:** 3

**Summary:**

This paper introduces ProofAug, a novel theorem-proving method that enhances the sample efficiency of proof synthesis by integrating automation tools at multiple granularity levels. Unlike prior approaches that use automation tools either selectively or at a single level, ProofAug applies fine-grained structure analysis to better leverage built-in tactics and automated theorem provers. Additionally, the method is designed as a plug-and-play module compatible with any tree-search algorithm, allowing the construction of an efficient recursive proving (ERP) module for further performance gains. Evaluated on the miniF2F-test benchmark using the deepseek-math-7b-base model and the Isabelle proof assistant, ProofAug achieves a new state-of-the-art (SOTA) performance, reaching a 66.0% cumulative pass rate with dataset curation (61.9% without) using only 2100 samples, aided by a mixed prompting strategy.

**Claims And Evidence:**

There are two major claims:

- ProofAug (based on proof structure analysis) improves pass rate given a relatively small sample budget.

- The cumulative pass rate when adopting a mixed prompting strategy achieves SOTA given a small sample budget.

The first claim is well demonstrated by detailed explanation of the system design and experiments. Table 2 shows clear differences between ProofAug and the DSP baseline when given the same amount of sample budget.

The second claim is weaker in my opinion. MiniF2F-test is a relatively small dataset, and a 0.1% improvement (66.0% vs 65.9%) is perhaps not significant enough to claim the crown (if not an error at all).

Moreover, since some MiniF2F-test problems are corrected (as reported in section 4.4), the performance of other existing approaches may also vary a bit. So it isn’t entirely fair to compare the numbers directly.

The small amount of sample budget does seem nice.

**Essential References Not Discussed:**

The references seem reasonable.

**Experimental Designs Or Analyses:**

The experimental designs and analyses look valid to me. I do find that the cumulative pass rate (with dataset curation) being the best is less convincing given that the test dataset is slightly modified / corrected and that the improvement is minimal. But the major part of this submission, namely the improvement in performance of vanilla ProofAug over DSP baseline is reasonable.

**Methods And Evaluation Criteria:**

The proposed methods make sense. It roughly follows the line of DSP but is much more carefully engineered to maximally exploit the capability of the built-in ATPs in Isabelle, aiming to balance the problem of hard conjectures and complicated draft (as described on page 2.). The evaluation criteria also makes sense. Pass rate with respect to sample budget is the standard metric in this area.

**Other Comments Or Suggestions:**

Section 3.3 mentions POETRY extensively and adopts part of its approach. Maybe add a high-level description in the very paragraph to briefly explain how the components work?

**Other Strengths And Weaknesses:**

I appreciate the amount of effort put into the careful engineering in this work. The presentation is also generally clear and the content is well-organized.

One thing I personally find counterintuitive is that in ProofAug the ATPs are essentially being tried first for smaller (i.e., innermost) subgoals and then larger ones. Usually one would expect that ATPs are better at solving low-level small goals and would struggle with larger ones. The fact that ProofAug works seems to suggest the opposite. This can be due to the fact that MiniF2F-test problems mainly suffer from the problem of “complicated draft”, where small goals are actually not suitable for ATPs, or it may just be accidental?

**Questions For Authors:**

Nothing in particular.

**Relation To Broader Scientific Literature:**

The approach proposed in this paper can potentially benefit provers other than Isabelle that have strong built-in ATPs.

**Theoretical Claims:**

The paper is mostly emprical.

---

> ### Author Rebuttal · Authors · 2025-03-31
>
> We thank the reviewer for the constructive feedback, especially the concise and accurate comments on our ProofAug method. Below is our response to your concerns:
>
> **Q1**: (On the major experimental claims: The improvement seems minimal, and correction of data might cause unfairness)
>
> **A1**: Thank you for the summary of our major claims from the experimental results. We apologize for the confusion caused by lack of explanation of the goals of the experimental comparisons and possibly inappropriate claims. We in fact want to claim the following three points through the experimental results:
> - Through ablation study, we show that ProofAug can improve pass rate given a relatively small sample budget. (as you mentioned)
> - When compared with previous Isabelle-based methods, our methods can achieve significant improvement on the pass rate, remarking a new SOTA and showing the overall superiority of the designs introduced in this work.
> - As to the comparison with methods based on other proof assistants, we agree with your opinion that a 0.1% improvement is not that significant. However, the main goal of our comparison is to show that, given that previous Isabelle-based methods have been largely outperformed by Lean-based counterparts, our method is still competitive with the state-of-the-art Lean-based methods under a far smaller sample budget.
>
> We will modify our claims to more clearly reflect these points in the revised version.
>
> As to the concern of the possible unfairness caused by our curation, we also report our results on the original Isabelle version of minif2f and show that our method is superior to other methods using Isabelle. The goal of curating the dataset is for the comparison with results evaluated on the Lean versions of minif2f. Our curation mainly consists of two types: 1. Fix typos. 2. Modify the type of the variables from Nat to Int for theorems that perform division on the variables (refer to our response to Q2 of Reviewer rCPC for detailed explanation of this type of curation). We have manually checked a large part of the Lean version of minif2f and find that no issues falling into these two types appear. Thus, we believe that using the evaluation result on our curated Isabelle version of minif2f to compare with the Lean version result is more fair.
>
>
> **Q2**: One thing I personally find counterintuitive is that in ProofAug the ATPs are essentially being tried first for smaller (i.e., innermost) subgoals and then larger ones. Usually one would expect that ATPs are better at solving low-level small goals and would struggle with larger ones. The fact that ProofAug works seems to suggest the opposite. This can be due to the fact that MiniF2F-test problems mainly suffer from the problem of “complicated draft”, where small goals are actually not suitable for ATPs, or it may just be accidental?
>
> **A2**: Our observation is that, for models comparable to or stronger than deepseek-math-7b-base, ATPs are actually better at solving low-level goals proposed by the models. We feel sorry that the specific 'complicated draft' example might have misled you. Note that the performance of sledgehammer on minif2f-test is only 20.9\%, which already indicates that the ATPs struggles to prove high-level competition-math theorems (since they are not designed to solve such problems). As to their performance on low-level model-proposed conjectures, during the experiments, we record at which stage ProofAug helps find a new proof, as you can observe in the results provided in the supplementary material together with the code. We classify the success stages into 3 types: 1. The initial proof succeeds; 2. The Maximal Compatible Semi-Proof succeeds after substituting all 'sorry's with <ATP>s; 3. Some coarse semi-proof succeeds after substituting all 'sorry's with <ATP>s. The rough observation is that , for deepseek-math-7b-base, type 2 success is more common than type 3, which means that the lowest-level conjectures it propose are at least quite 'useful' for the ATPs.
>
>
> **Q3**: Section 3.3 mentions POETRY extensively and adopts part of its approach. Maybe add a high-level description in the very paragraph to briefly explain how the components work?
>
> **A3**: Thank you for the suggestion. We thought that POETRY appears in the Table 1 in the preliminary section so we chose not to describe it extendedly in Section 3.3. We apologize for the confusion and will add a high-level description to briefly explain how the components work in the revised version.
>
>
> Finally, we thank the reviewer for the time and consideration. We hope that our responses have addressed your concerns.

---

### Official Review · Reviewer_rCPC · 2025-03-19

**Overall Recommendation:** 4

**Summary:**

The paper proposes ProofAug, a method for achieving efficient neural theorem proving by combining LLMs with automated theorem proving (ATP). The paper conducts extensive experiments comparing ProofAug with baseline methods, categorizing them by different proof styles.

**Claims And Evidence:**

Yes. The paper addresses the trade-off between generating hard conjectures and producing overly complicated drafts, particularly in the DSP framework. It claims to propose a solution that effectively balances the two. The proposed algorithm demonstrates its effectiveness through experiments, including ablation studies, which show that ProofAug improves performance over DSP by a significant margin.

**Essential References Not Discussed:**

None.

**Experimental Designs Or Analyses:**

The paper begins by categorizing different baseline approaches and then evaluates the performance of the proposed method, ensuring that the sampling budget aligns with baseline methods for a fair comparison. The evaluation results on the miniF2F test dataset show a clear improvement in performance. The paper also conducts ablation studies on the Efficient Recursive Proving (ERP) module and mixed proof strategies to validate the necessity of each component.

**Methods And Evaluation Criteria:**

The method is evaluated on the miniF2F test dataset (in Isabelle), which is a widely accepted benchmark for theorem proving. The dataset includes theorems of varying difficulty levels, making it a reasonable choice for evaluating theorem-proving methods.

**Other Comments Or Suggestions:**

I don't have other comments.

**Other Strengths And Weaknesses:**

Strengthes and weaknesses are fully discussed in other sections.

**Questions For Authors:**

1. Regarding lines 261-263 in Section 3.2: Instead of replacing the outer block with sorry, have the authors considered generating more fine-grained drafts from the exact inner theorem containing sorry? It seems counterintuitive that an advanced Isabelle tactic would be more effective on a coarse theorem than on a finer one.

2. Regarding the curation of the miniF2F-test dataset: How are the incorrect theorems classified as incorrect? Are they unprovable in principle, or are there formalization errors that make them trivially provable? Providing concrete examples would improve clarity.

3. Why was Isabelle selected over Lean for evaluation? Is it due to the authors’ familiarity with the proof language, or does Isabelle have an advantage in terms of automation tactics or ATP integration?

**Relation To Broader Scientific Literature:**

The paper reviews existing work in neural theorem proving, categorizing approaches into whole-proof generation, proof-step generation, and hybrid methods. By building on these existing paradigms and introducing a framework that improves performance through fine-grained proof structure analysis, the paper advances the state of the art in this field. Additionally, the integration of ProofAug into tree-search algorithms aligns with recent advances in proof synthesis and recursive theorem proving.

**Theoretical Claims:**

The paper does not contain theoretical claims.

---

> ### Author Rebuttal · Authors · 2025-03-31
>
> We thank the reviewer for the clear and constructive feedback. As to your questions, we address them as follows:
>
> **Q1**: Regarding lines 261-263 in Section 3.2: Instead of replacing the outer block with sorry, have the authors considered generating more fine-grained drafts from the exact inner theorem containing sorry? It seems counterintuitive that an advanced Isabelle tactic would be more effective on a coarse theorem than on a finer one.
>
> **A1**: Yes! We think what you describe is exactly our ERP module introduced in Section 3.3. We apologize for the unclearness in our description of ERP module. A clearer version of description of ERP is that, for each conjecture (inner theorem) in the Maximal Compatible Semi-Proof that <ATP> fails to prove, we ask the model to generate a new draft for it, given the corresponding proof state in the comment in hope that this information can help it draw a more fine-grained draft. You can refer to line 290-300 in Algorithm 2 and our code (the proof_aug_1dp method) for more details.
>
> In the revised version, we will modify the line 311-315 in Section 3.3 to 'In practice, to simplify the algorithm and suppress the sampling cost wasted in low-level details, we only try $\verb|<ATP>|$ and add new nodes for failed conjectures belonging to the original MCSP' to make our description clearer.
>
>
> **Q2**: Regarding the curation of the miniF2F-test dataset: How are the incorrect theorems classified as incorrect? Are they unprovable in principle, or are there formalization errors that make them trivially provable? Providing concrete examples would improve clarity.
>
> **A2**: We only checked the theorems that cannot get proved in our initial experiments, so we think our curation does not include theorems that are made trivially provable due to formalization errors. There are mainly two types of our curation: 1.Fix Typos. Statements with typos in variable names or numbers are mostly completely wrong and unprovable. 2. Modifying the type of the variables from Nat to Int for theorems that perform division on the variables. This is important because, for example, the operation "(2::Nat) - (3::Nat)" is valid and expected to be "(0::Nat)" in Isabelle. Although sometimes the original minif2f statement can still be proved in the sense that the resulted 0 also satisfies the constraints in the problem, but it is not expressing what the original problem intends. Below is an example:
> ```html
> Differences:
> --- Original
> +++ Fixed
> @@ -1,8 +1,8 @@
>  theorem imo_2001_p6:
> -  fixes a b c d ::nat
> +  fixes a b c d ::int
>    assumes "0 < a \<and> 0 < b \<and> 0 < c \<and> 0 < d"
>      and "d < c"
>      and "c < b"
>      and "b < a"
> -    and "a * c + b * d = (b + d + a - c) * (b + d - a + c)"
> +    and "a * c + b * d = (a + b - c + d) * (- a + b + c + d)"
>    shows "\<not> prime (a * b + c * d)"
> HTML diff saved to xf_diffs/imo_2001_6_diff.html
> ```
> The "a * c + b * d = (b + d + a - c) * (b + d - a + c)" constraint does not express what it intends in the original statement when $a > b+d$ or $c > b +d+a$. We will also provide several typical examples in our revised version of the paper.
>
>
> **Q3**: Why was Isabelle selected over Lean for evaluation? Is it due to the authors’ familiarity with the proof language, or does Isabelle have an advantage in terms of automation tactics or ATP integration?
>
> **A3**: Mostly the latter one. Isabelle/HOL has the most powerful proof automation across all proof assistants, and the sledgehammer is a built-in tool for Isabelle that have been developed for years. In comparison, Lean does not include an built-in SMT implementation or access to external ATP/SMTs with it, and the related community projects were mostly still in beta during our work.
> As a result, it is most appropriate to try out our method in Isabelle/HOL first to verify its effectiveness, or in other words, to obtain a 'proof of concept'.
> You may also refer to our response to Q5 of Reviewer vgEx for the obstacles we are facing when trying to implement ProofAug on Lean.
>
>
> Again, we appreciate your constructive feedback and will make the necessary changes in our revised version. We hope that our responses have addressed your concerns and clarified the points you raised. Thank you for your time and consideration.

---

### Decision · Program_Chairs · 2025-05-01

**Decision:**

Accept (poster)

**Comment:**

The paper presents ProofAug, a novel approach to neural theorem proving that enhances sample efficiency by integrating large language models (LLMs) with traditional automation tools at multiple granularity levels. The method is evaluated on the miniF2F-test benchmark, achieving a new state-of-the-art (SOTA) performance with a 66.0% cumulative pass rate after dataset curation, surpassing previous results.

(+) The method shows significant improvements in sample efficiency, making it a valuable contribution to the field of neural theorem proving.
(-/=) The field of neural theorem proving is moving so fast, Kimina-Prover is already at 80%+ on miniF2F.
(+) The paper is very clear and provides ample ablations.

Comments to Authors:
* Consider providing more quantitative evidence for the limitations of the DSP framework as discussed in the introduction.
* Clarify the curation process of the miniF2F-test dataset and ensure that comparisons with baseline methods are fair and transparent.
* Explore the implementation and evaluation of ProofAug on other proof assistants, such as Lean, to demonstrate its generalizability and potential impact across different systems.

Overall, the paper presents a valuable contribution to the field and is recommended for acceptance at ICML 2025.